# CO-MOT: Boosting End-to-end Transformer-based Multi-Object Tracking via Coopetition Label Assignment and Shadow Sets

## Abstract

Existing end-to-end Multi-Object Tracking (e2e-MOT) methods have not sur-passed non-end-to-end tracking-by-detection methods. One potential reason is its label assignment strategy during training that consistently binds the tracked objects with tracking queries and then assigns the few newborns to detection queries. With one-to-one bipartite matching, such an assignment will yield an unbalanced training, *i.e.*, scarce positive samples for detection queries, especially for an en-closed scene, as the majority of the newborns come on stage at the beginning of videos. Thus, e2e-MOT will be easier to yield a tracking terminal without renewal or re-initialization, compared to other tracking-by-detection methods. To allevi-ate this problem, we present Co-MOT, a simple and effective method to facilitate e2e-MOT by a novel coopetition label assignment with a shadow concept. Specif-ically, we add tracked objects to the matching targets for detection queries when performing the label assignment for training the intermediate decoders. For query initialization, we expand each query by a set of shadow counterparts with limited disturbance to itself. With extensive ablations, Co-MOT achieves superior perfor-mance without extra costs, *e.g.*, 69.4% HOTA on DanceTrack and 52.8% TETA on BDD100K. Impressively, Co-MOT only requires 38% FLOPs of MOTRv2 to attain a similar performance, resulting in the $1.4\times$ faster inference speed. Codes are attached for re-implementation.

## 1 Introduction

Multi-Object tracking (MOT) is traditionally tackled by a series of tasks, *e.g.*, object detection( Zou et al. (2023); Tan et al. (2020); Redmon et al. (2016); Ge et al. (2021)), appearance Re-ID( Zheng et al. (2016); Li et al. (2018); Bertinetto et al. (2016)), motion prediction( Lefèvre et al. (2014); Welch et al. (1995)), and temporal association( Kuhn (1955)). The sparkling advantage of this paradigm is task decomposition, leading to an optimal solution for each task. However, it lacks global optimization for the whole pipeline.

Recently, end-to-end Multi-Object Tracking (e2e-MOT) via Transformer such as MOTR( Zeng et al. (2022)) and TrackFormer( Meinhardt et al. (2022)) has emerged, which performs detection and tracking simultaneously in unified transformer decoders. Specifically, tracking queries realize iden-tity tracking by recurrent attention over time. Meanwhile, detection queries discover newborns in each new arriving frame, excluding previously tracked objects, due to a Tracking Aware Label As-signment (TALA) during training. However, we observe an inferior performance for e2e-MOT due to poor detection, as it always yields a tracking terminal, shown in Figure 1. MOTRv2( Zhang et al. (2023)) consents to this conclusion, which bootstraps performance by a pre-trained YOLOX( Ge et al. (2021)) detector, but the detector will bring extra overhead to deployment.

In this paper, we present a novel viewpoint for addressing the above limitations of e2e-MOT: **de-tection queries are exclusive but also conducive to tracking queries**. To this end, we develop a COopetition Label Assignment (COLA) for training tracking and detection queries. Except for the last Transformer decoder remaining the competition strategy to avoid trajectory redundancy, we allow the previously tracked objects to be reassigned to the detection queries in the intermediate decoders. Due to the self-attention between all the queries, detection queries will be complementary

to tracking queries with the same identity, resulting in feature augmentation for tracking objects with significant appearance variance. Thus, the tracking terminal problem will be alleviated.

Besides TALA, another drawback in Transformer-based detection as well as tracking is one-to-one bipartite matching used, which cannot produce sufficient positive samples, as denoted by Co-DETR( Zong et al. (2023)) and HDETR( Jia et al. (2023)) that introduces one-to-many assignment to overcome this limitation. Differing from these remedies with one-to-many auxiliary training, we develop a **one-to-set matching strategy with a novel shadow concept**, where each individual query is augmented with multiple shadow queries by adding limited disturbance to itself, so as to ease the one-to-set optimization. The set of shadow queries endows Co-MOT with discriminative training by optimizing the most challenging query in the set with the maximal cost. Hence, the generalization ability will be enhanced.

We evaluate our proposed method on multiple MOT benchmarks, including DanceTrack( Sun et al. (2022)), BDD100K( Yu et al. (2020)) and MOT17( Milan et al. (2016)), and achieve superior performance. The contributions of this work are threefold: i) we introduce a coopetition label assignment for training tracking and detection queries for e2e-MOT with high efficiency; ii) we develop a one-to-set matching strategy with a novel shadow concept to address the hungry for positive training samples and enhance generalization ability; iii) Our approach achieves superior performance on multiple benchmarks, while it functions as an efficient tool to boosting the performance of end-to-end Transformer-based MOT.

## 2 RELATED WORKS

**Tracking by detection**: Most tracking algorithms are based on the two-stage pipeline of tracking-by-detection: Firstly, a detection network is used to detect the location of targets, and then an association algorithm is used to link the targets across different frames. However, the performance of this method is greatly dependent on the quality of the detection. SORT( Bewley et al. (2016)) is a widely used object tracking algorithm that utilizes a framework based on Kalman filters( Welch et al. (1995)) and the Hungarian algorithm( Kuhn (1955)); After, new methods are proposed, *e.g.*, Deep SORT( Wojke et al. (2017)), JDE( Wang et al. (2020)), FairMOT( Zhang et al. (2021)), GTR( Zhou et al. (2022)), TransTrack( Sun et al. (2020)), QuasiDense( Pang et al. (2021)), TraDeS( Wu et al. (2021)), CenterTrack( Stone et al. (2000)), Tracktor++( Bergmann et al. (2019)); Recently, Byte-Track( Zhang et al. (2022b)), OC-SORT( Cao et al. (2023)), MT_IOT( Yan et al. (2022b)), Strong-sort ( Du et al. (2023)), BoT-SORT( Aharon et al. (2022)) are proposed, that have further improved the tracking performance by introducing the strategy of matching with low-confidence detection boxes. While these methods show improved performance, they often require significant parameter tuning and may be sensitive to changes in the data distribution. Additionally, some approaches may require more advanced techniques such as domain adaptation or feature alignment to effectively handle domain shift issues.

**End-to-end tracking**: With the recent success of Transformer in various computer vision tasks, several end-to-end object tracking algorithms using Transformer encoder and decoder modules are proposed, such as MOTR and TrackFormer. These approaches demonstrate promising results in object tracking by directly learning the associations between object states across time steps. MOTRv2 introduces the use of pre-detected anchor boxes from a YOLOX detector to indirectly achieve state-of-the-art performance in multi-object tracking.

**One-to-many label assignment**: DETR( Carion et al. (2020)), being a pioneer in employing transformers for computer vision, utilizes a one-to-one label assignment strategy to achieve end-to-end object detection. During training, DETR leverages Hungarian matching to compute the global matching cost and thereby assigns each ground-truth box to a unique positive sample. Researchers shifte focus towards enhancing the performance of DETR, with most efforts concentrated on developing new label assignment techniques. For example, DN-DETR( Li et al. (2022a)) building on Deformable DETR( Zhu et al. (2020)), breaks away from the traditional one-to-one matching strategy by introducing noisy ground-truth boxes during training. DINO( Zhang et al. (2022a)) builds upon the successes of DN-DETR( Li et al. (2022a)) and DAB-DETR( Liu et al. (2022)) to achieve an even higher detection performance, putting it at the forefront of current research. Group-DETR( Chen et al. (2023)), H-DETR( Jia et al. (2023)), CO-DETR( Zong et al. (2022)) start using the concept of groups to accelerate convergence.

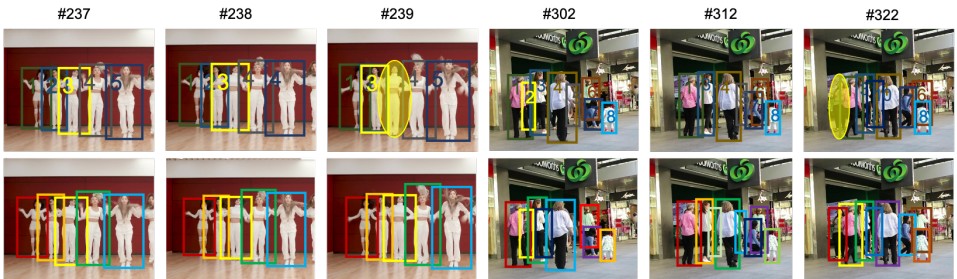

Figure 1: Visualization of tracking results in DanceTrack0073 and MOT17-09 videos. The first row displays the tracking results from MOTR, where all individuals can be correctly initialized at the beginning (#237 and #302). However, heavy occlusion appears in the middle frames (#238 and #312), resulting in inaccurate detection (indicated by yellow boxes). The tracking of yellow targets finally terminates in #239 and #322 frames. The second row shows MOTR's detection results, in which tracking queries are removed during the inference process. Targets in different frames are accurately detected.

Table 1: The detection performance (mAP) of MOTR (v2) on DanceTrack validation dataset. ✓means whether the tracking queries are used in the training or inference phase. All the decoded boxes of both tracking if applicable and detection queries are treated as detection boxes for evaluation on mAP. We separately evaluate the detection performance for six decoders. For analysis, please refer to the motivation section.

| | methed | Training | Inference | 1 | 2 | 3 | 4 | 5 | 6 |
|---|---|---|---|---|---|---|---|---|---|
| (a) | MOTR | ✓ | ✓ | 41.4 | 42.4 | 42.5 | 42.5 | 42.5 | 42.5 |
| (b) | MOTR | ✓ | | 56.8 | 60.1 | 60.5 | 60.5 | 60.6 | 60.6 |
| (c) | MOTR | | | 57.3 | 62.2 | 62.9 | 63.0 | 63.0 | 63.0 |
| (d) | MOTRv2 | ✓ | ✓ | 67.9 | 70.2 | 70.6 | 70.7 | 70.7 | 70.7 |
| (e) | MOTRv2 | ✓ | | 71.9 | 72.1 | 72.1 | 72.1 | 72.1 | 72.1 |

## 3 METHOD

### 3.1 MOTIVATION

To explore the shortcomings of current end-to-end methods in tracking, we conduct an in-depth study of the effectiveness on DanceTrack validation and MOT17 test dataset by analyzing MOTR, which is one of the earliest proposed end-to-end multiple-object tracking methods. In Figure 1, we show MOTR's tracking results in some frames of video, *e.g.*, DanceTrack0073 and MOT17-09. In the left three columns of the first row, the 3rd person (in the yellow box) is tracked normally in #237 image. However, in #238 image, due to an inaccurate detection, the bounding box is not accurately placed around that person (the box is too large to include a person on the left side). In #239 image, the tracking is completely wrong and associated with the 2nd person instead. In the right three columns of the first row, the 2nd person (in the yellow box) is successfully detected and tracked in #302 image. However, in #312 image, this person is occluded by other people. When the person appears again in #322 image, she is not successfully tracked or even detected. To determine whether the tracking failure is caused by the detection or association of MOTR, we visualized MOTR's detection results in the second row. We remove the tracking queries during inference, and the visualization shows that all persons are accurately detected. This demonstrates that the detection will deteriorate due to the nearby tracked objects, though TALA used in training ensures that the detection with the same identity of tracked objects will be suppressed.

We further provide quantitative results of how the queries affect each other in Table 1. All the decoded boxes of both tracking and detection queries are treated as detection boxes so that they can be evaluated by the mAP metric commonly used for object detection. We can see from the table that the vanilla MOTR (a) has a low mAP 42.5%, but it increases by 18.1% (42.5% vs 60.6%) when removing tracking queries during inference (b). Then we retrain MOTR as a sole detection task

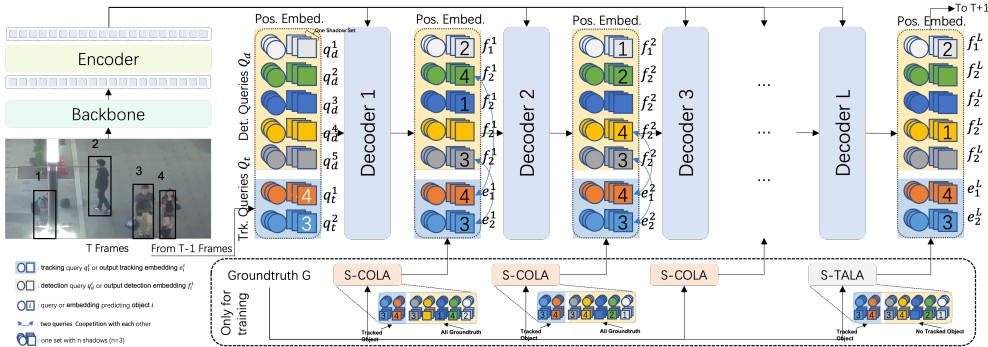

Figure 2: The CO-MOT framework includes a CNN-based backbone network for extracting image features, a deformable encoder for encoding image features, and a deformable decoder that uses self-attention and cross-attention mechanisms to generate output embeddings with bounding box and class information. The queries in the framework use set queries as units, with each set containing multiple shadows that jointly predict the same target. Detection queries and tracking queries are used for detecting new targets and tracking existing ones, respectively. To train CO-MOT, S-COLA and S-TALA are proposed for training only.

by removing tracking queries (c) and mAP further increases to 66.1% (+5.5%). That means the DETR-style MOT model has a sparking capability of detection but still struggles with the temporal association of varied appearance, which is the crucial factor of MOT.

We also observe an excellent detection performance (70.7%) for MOTRv2, which introduces a pretrained YOLOX detector. Removing tracking queries during inference brings a slight improvement (1.4%) for mAP, which means MOTRv2 has almost addressed the poor detection issue with high-quality detection prior from YOLOX. **However, the introduced YOLOX brings extra computational burden, unfriendly to deployment. In contrast, we intend to endow the end-to-end MOT model with its own powerful detection capability, rather than introducing any extra pretrained detector.**

### 3.2 TRACKING AWARE LABEL ASSIGNMENT

Here we revisit the Tracking Aware Label Assignment (TALA) used to train end-to-end Transformers such as MOTR and TrackFormer for MOT. At the moment $t - 1$, $N$ queries are categorized to two types: $N_T$ tracking queries $Q_t = \{q_t^1, ..., q_t^{N_T}\}$ and $N_D$ detection queries $Q_d = \{q_d^1, ..., q_d^{N_D}\}$, where $N = N_T + N_D$. All the queries will self-attend each other and then cross-attend the image feature tokens via $L$ decoders, and the output embeddings of the $l$-th decoder are denoted as $E^l = \{e_1^l, ..., e_{N_T}^l\}$ and $F^l = \{f_1^l, ..., f_{N_D}^l\}$. At the moment $t$, there are $M_G$ ground truth boxes. Among them, $M_T$ previously tracked objects, denoted as $\hat{E} = \{\hat{e}_1, ..., \hat{e}_{M_T}\}$, are assigned to $N_T$ tracking queries, where $M_T \leq N_T$ as some objects disappear. Formally, $j$-th tracking embedding $e_j^l$ will be assigned to the same identity with the previous timestamp if still alive at this moment, otherwise zero (disappearing). Besides, $M_D$ newborn objects, denoted as $\hat{F} = \{\hat{f}_1, ..., \hat{f}_{M_D}\}$, are assigned to $N_D$ detection queries. Specifically, the Hungarian matching algorithm is used to find the optimal pairing between $F^i$ and $\hat{F}$ for each decoder, by a cost function $(L_m = L_f(c) + L_1(b) + L_g(b) \in R^{N_D * M_G})$, that takes into account the class scores and box overlapping. Where $L_f(c)$ represents the focal loss for classification, $L_1(b)$ represents the $L_1$ cost of the bounding box, and $L_g(b)$ represents the Generalized Intersection over Union cost.

### 3.3 OVERALL ARCHITECTURE

The entire CO-MOT framework is illustrated in Figure 2. During the forward process, the features of an image in a video are extracted by the backbone and fed into the deformable encoder to aggregate information. Finally, together with the detection and tracking queries, they are used as the inputs of the $L$ layer decoders ($L = 6$ in this paper by default) to detect new targets or track the already tracked targets. It is worth noting that queries contain $(N_T + N_D) * N_S$ position ($\mathbb{P} \in \mathbb{R}^4$) and

embedding ($\mathbb{E} \in \mathbb{R}^{256}$) as we use deformable attention. Here $N_S$ is the number of shadow queries for each set, and we will introduce the shadow set concept in the following section. All the queries predict $(N_T + N_D) * N_S$ target boxes, where $N_S$ queries in a set jointly predict the same target. To train CO-MOT, we employ the COLA and TALA on the different decoders, along with the one-to-set label assignment strategy.

### 3.4 COOPETITION LABEL ASSIGNMENT

Unlike TALA, which only assigns newborn objects to detection queries, we advocate a novel COopetition Label Assignment (COLA). Specifically, we assign $M_T$ tracked objects to detection queries as well in the intermediate decoders, *i.e.*, $l < L$, which is illustrated in Figure 2. As shown in the output of the first decoder, the track queries continue to track the 3rd and 4th person. The detection queries not only detect the 1st and 2nd newborns but also detect the 3rd and 4th people. Note that we remain the competition assignment for the $L$-th decoder to avoid trajectory redundancy during inference. Thanks to the self-attention used between tracking and detection queries, detection queries with the same identity can enhance the representation of the corresponding tracking queries (*e.g.* grey 3rd helps blue 3rd).

### 3.5 SHADOW SET

In densely crowded scenes, objects can be lost or mistakenly tracked to other objects due to minor bounding box fluctuations. We conjecture that one query for one object is sensitive to prediction noises. Inspired by previous works such as Group-DETR and H-DETR, we propose the one-to-set label assignment strategy for multi-object tracking, which is significantly different from the one-to-many manner. During the tracking, an object is no longer tracked by a single query but by a set of queries, where each member of the set acts as a shadow of each other. Tracking queries are rewritten as $Q_t = \{\{q_t^{1,i}\}_{i=1}^{N_S}, ..., \{q_t^{N_T,i}\}_{i=1}^{N_S}\}$ and detection queries are rewritten as $Q_d = \{\{q_d^{1,i}\}_{i=1}^{N_S}, ..., \{q_d^{N_D,i}\}_{i=1}^{N_S}\}$. The total number of queries is $N * N_S$. When a particular query in the set tracks the object incorrectly, the other shadows in the same set help it continue tracking the object. In the experiments, this strategy prove effective in improving tracking accuracy and reducing tracking failures in dense and complex scenes.

**Initialization.** $P^{i,j} \in \mathbb{R}^4$ and $X^{i,j} \in \mathbb{R}^{256}$, which represents position and embedding of the $j$-th shadow query in the $i$-th set, are initialized, which significantly affects the convergence and the final performance. In this paper, we explore three initialization approaches: i) $I_{rand}$: random initialization; ii) $I_{copy}$: initializing all shadows in the same set with one learnable vector, *i.e.*, $P^{i,j} = P^i$ and $X^{i,j} = X^i$, where $P^i$ and $X^i$ are learnable embeddings with random initialization; iii) $I_{noise}$: adding Gaussian noises $\mathcal{N}(0, \sigma_p)$ and $\mathcal{N}(0, \sigma_x)$ to $P^{i,j}$ and $X^{i,j}$, respectively, in the previous approach. In the experiment, we set $\sigma_p$ and $\sigma_x$ to 1e-6. Although the variance between each shadow in the same set is subtle after initialization, it expands to 1e-2 at the end of training. The last approach provides the similarity for helping optimization and diversity to improve tracking performance.

**Training.** We propose a shadow-based label assignment method (S-COLA or S-TALA) to ensure that all objects within a set are matched to the same ground truth object. Take S-COLA as an example, we treat the set as a whole, and select one of them as a representative based on criteria to participate in subsequent matching. Specifically, for tracking queries $Q_t$, the tracked target in the previous frame is selected to match with the whole set; For detection queries $Q_d$, we first calculate the cost function ($L_{sm} \in R^{N_D * N_S * M_G}$) of all detection queries with respect to all ground truth. We then select the representative query by a strategy $\lambda$ (*e.g.*, Mean, Min, and Max) for each set, resulting in $L_m = \lambda(L_{sm}) \in R^{N_D * M_G}$. $L_m$ is then used as an input for Hungarian matching to obtain the matching results between the sets and newborns. Finally, the other shadows within the same set share the representative's matching result.

**Inference.** We determine whether the $i$-th shadow set tracks an object by the confidence score of the selected representative. Here we adopt a different strategy $\phi$ (*e.g.*, Mean, Min, and Max) for representative sampling. When the score of the representative is higher than a certain threshold $\tau$, we select the box and score predictions of the shadow with the highest score as the tracking outputs and feed the entire set to the next frame for subsequent tracking. Sets that do not capture any object will be discarded.

## 4 EXPERIMENT

Table 2: Comparison to state-of-the-art methods on different dataset. Please pay more attention to the metrics with blue.

(a) Comparison to existing methods on the Dance-Track test set. "*", "+" respectively represent the use of DAB-Deformable backbone and joint training with CrowdHuman. Best results of end-to-end methods are marked in bold.

|  | HOTA | DetA | AssA | MOTA | IDF1 |
|---|---|---|---|---|---|
| Non-End-to-end | | | | | |
| ByteTrack | 47.7 | 71.0 | 32.1 | 89.6 | 53.9 |
| QDTrack | 54.2 | 80.1 | 36.8 | 87.7 | 50.4 |
| OC-SORT | 55.1 | 80.3 | 38.3 | 92.0 | 54.6 |
| TransTrack | 45.5 | 75.9 | 27.5 | 88.4 | 45.2 |
| MOTRv2+ | 69.9 | 83.0 | 59.0 | 91.9 | 71.7 |
| End-to-end | | | | | |
| MOTR | 54.2 | 73.5 | 40.2 | 79.7 | 51.5 |
| MeMOTR | 63.4 | 77.0 | 52.3 | 85.4 | 65.5 |
| MeMOTR* | 68.5 | 80.5 | 58.4 | 89.9 | 71.2 |
| CO-MOT | 65.3 | 80.1 | 53.5 | 89.3 | 66.5 |
| CO-MOT+ | **69.4** | **82.1** | **58.9** | **91.2** | **71.9** |

(b) Comparison to existing methods on the BDD100K validation set.

|  | TETA | LocA | AssocA | ClsA |
|---|---|---|---|---|
| DeepSORT | 48.0 | 46.4 | 46.7 | 51.0 |
| QDTrack | 47.8 | 45.8 | 48.5 | 49.2 |
| TETer | 50.8 | 47.2 | 52.9 | 52.4 |
| MOTR | 50.7 | 35.8 | 51.0 | / |
| MOTRv2 | **54.9** | **49.5** | 51.9 | 63.1 |
| CO-MOT(ours) | 52.8 | 38.7 | **56.2** | **63.6** |

(c) Comparison to existing methods on the MOT17 test dataset. Best results of end-to-end methods are marked in bold.

|  | HOTA | DetA | AssA | MOTA | IDF1 |
|---|---|---|---|---|---|
| Non-End-to-end | | | | | |
| Tracktor++ | 44.8 | 44.9 | 45.1 | 53.5 | 52.3 |
| CenterTrack | 52.2 | 53.8 | 51.0 | 67.8 | 64.7 |
| TraDeS | 52.7 | 55.2 | 50.8 | 69.1 | 63.9 |
| QuasiDense | 53.9 | 55.6 | 52.7 | 68.7 | 66.3 |
| TransTrack | 54.1 | 61.6 | 47.9 | 74.5 | 63.9 |
| GTR | 59.1 | 57.0 | 61.6 | 71.5 | 75.3 |
| FairMOT | 59.3 | 60.9 | 58.0 | 73.7 | 72.3 |
| CorrTracker | 60.7 | 62.9 | 58.9 | 76.5 | 73.6 |
| Unicorn | 61.7 | / | / | 77.2 | 75.5 |
| GRTU | 62.0 | 62.1 | 62.1 | 74.9 | 75.0 |
| MAATrack | 62.0 | 64.2 | 60.2 | 79.4 | 75.9 |
| ByteTrack | 63.1 | 64.5 | 62.0 | 80.3 | 77.3 |
| OC-SORT | 63.2 | / | 63.2 | 78.0 | 77.5 |
| QDTrack | 63.5 | 62.6 | 64.5 | 77.5 | 78.7 |
| BoT-SORT | 64.6 | / | / | 80.6 | 79.5 |
| Deep OC-SORT | 64.9 | / | / | 80.6 | 79.4 |
| P3AFormer | / | / | / | 81.2 | 78.1 |
| MOTRv2 | 62.0 | 63.8 | 60.6 | 78.6 | 75.0 |
| End-to-end | | | | | |
| TrackFormer | / | / | / | 65.0 | 63.9 |
| MOTR | 57.8 | **60.3** | 55.7 | **73.4** | 68.6 |
| MeMOT | 56.9 | / | 55.2 | 72.5 | 69.0 |
| MeMOTR | 58.8 | 59.6 | 58.4 | 72.8 | 71.5 |
| CO-MOT | **60.1** | 59.5 | **60.6** | 72.6 | **72.7** |

### 4.1 DATASETS AND METRICS

**Datasets.** We validate the effectiveness of our approach on different datasets, including DanceTrack, MOT17, and BDD100K. Each dataset has its unique characteristics and challenges.

The DanceTrack dataset is used for multi-object tracking of dancers and provides high-quality annotations of dancer motion trajectories. This dataset is known for its significant difficulties such as fast object motion, object similar appearances

The MOT17 dataset is a commonly used multi-object tracking dataset, and each video contains a large number of objects. The challenges of this dataset include high object density, long-period occlusions, varied object sizes, dynamic camera poses, and so on.

The BDD100K dataset is a large-scale autonomous driving scene recognition dataset that is used for scene understanding in autonomous driving systems. This dataset provides multiple object categories, such as cars, pedestrians, etc. The challenges of this dataset include rapidly changing traffic and road conditions, diverse weather conditions, and lighting changes.

**Metrics.** To evaluate our method, we use the Higher Order Tracking Accuracy (HOTA) metric ( et al. (2020)), which is a higher-order metric for multi-object tracking. Meantime We analyze the contributions of Detection Accuracy (DetA), Association Accuracy (AssA), Multiple-Object Tracking Accuracy (MOTA), Identity Switches (IDS), and Identity F1 Score (IDF1). For BDD100K, to better evaluate the performance of multi-class and multi-object tracking, we use the Tracking Every Thing Accuracy (TETA)( Li et al. (2022b)), Localization Accuracy (LocA), Association Accuracy (AssocA), and Classification Accuracy(ClsA) metrics.

Table 3: Ablation studies of our proposed CO-MOT on the DanceTrack validation set. Please pay more attention to the metrics with blue.

(a) Ablation study on individual CO-MOT components. As components are added, the tracking performance improves gradually.

| | COLA | Shadow | HOTA | DetA | AssA | MOTA | IDF1 |
|---|---|---|---|---|---|---|---|
| (a) | | | 56.4 | 71.8 | 44.6 | 79.8 | 57.5 |
| (b) | ✓ | | 60.2 | 73.2 | 49.7 | 81.8 | 62.4 |
| (c) | | ✓ | 59.0 | 72.6 | 48.2 | 80.9 | 59.6 |
| (d) | ✓ | ✓ | 61.8 | 73.5 | 52.2 | 81.7 | 63.3 |

(b) Effect of different $\lambda$ and $\phi$ combinations.

| $\lambda$ | max | | | mean | | | min | | |
|---|---|---|---|---|---|---|---|---|---|
| $\phi$ | min | mean | max | min | mean | max | min | mean | max |
| HOTA | 57.6 | 56.4 | 55.1 | 56.7 | 55.2 | 52.0 | 57.5 | 55.9 | 51.5 |
| DetA | 70.7 | 69.3 | 65.4 | 70.6 | 66.5 | 59.0 | 70.8 | 66.4 | 59.3 |
| AssA | 47.3 | 46.1 | 46.7 | 45.9 | 46.1 | 46.1 | 46.8 | 47.2 | 45.0 |

(c) Effect of initialization methods for shadow queries $I_m$ and number of shadows $N_S$ on the DanceTrack validation set.

| | | HOTA | DetA | AssA |
|---|---|---|---|---|
| | $I_{copy}$ | 60.6 | 73.9 | 50.0 |
| $I_m$ | $I_{noise}$ | 61.5 | 73.1 | 51.9 |
| | $I_{rand}$ | 59.6 | 73.2 | 48.9 |
| | 1 | 60.2 | 73.2 | 49.7 |
| | 2 | 61.5 | 73.1 | 51.9 |
| $N_S$ | 3 | 61.8 | 73.5 | 52.2 |
| | 4 | 60.8 | 73.8 | 50.3 |
| | 5 | 60.2 | 72.2 | 50.5 |
| | 6 | 59.1 | 70.5 | 49.8 |

## 4.2 IMPLEMENTATION DETAILS

Our proposed label assignment and shadow concept can be applied to any e2e-MOT method. For simplicity, we conduct all the experiments on MOTR. It uses ResNet50 as the backbone to extract image features and uses a Deformable encoder and Deformable decoder to aggregate features and predict object boxes and categories. We also use the data augmentation methods employed in MOTR, including randomly clipping and temporally flipping a video segment. To sample a video segment for training, we use a fixed sampling length of 5 and a sampling interval of 10. The dropout ratio in attention is zero. We train all experiments on 8 V100-16G GPUs, with a batch size of 1 per GPU. For DanceTrack and BDD100k, we train the model for 20 epochs with an initial learning rate of 2e-4 and reduce the learning rate by a factor of 10 every eight epochs. For MOT17, we train the model for 200 epochs, with the learning rate reduced by a factor of 10 every 80 epochs. We use 300 initial queries due to the large number of targets to be tracked.

## 4.3 COMPARISON WITH STATE-OF-THE-ART METHODS

**DanceTrack.** Our method presents promising results on the DanceTrack test set, as evidenced by Table 2a. As shown in the original paper( Gao & Wang (2023)), the backbone used by MeMOTR is the original version of Deformable DETR, which is the same as the one we use, while that of MeMOTR* is DAB-Deformable-DETR( Liu et al. (2022)). Without bells and whistles, our method achieve an impressive HOTA score of 69.4%. In comparison with tracking-by-detection methods, such as QDTrack( Fischer et al. (2022)), OC-SORT, our approach stands out with a significant improvement in a variety of tracking metrics. For example, compared to OC-SORT, CO-MOT improves HOTA, and AssA by 10.2%, and 15.2%, respectively. Our approach can avoid tedious parameter adjustments and ad hoc fusion of two independent detection and tracking modules. It realizes automatic learning of data distribution and global optimization objectives. Compared to other end-to-end methods, such as MOTR, MeMOTR, CO-MOT outperforms them by a remarkable margin (*e.g.,* 11,1% improvement on HOTA compared to MOTR, 1.9% compared to MeMOTR). **Note that CO-MOT$^+$ has a comparable performance with MOTRv2 which introduces an extra pre-trained YOLOX detector to MOTR. Both apply joint training on CrowdHuman.**

**BDD100K.** Table 2b shows the results of different tracking methods on the BDD100K validation set. To better evaluate the multi-category tracking performance, we adopt the latest evaluation metric TETA, which combines multiple factors such as localization, association and classification. Compared with DeepSORT, QDTrack, and TETer( Li et al. (2022b)), MOTR, although the LocA was considerably lower, we achieve superior performance on TETA with an improvement of 2% (52.8% vs 50.8%), which is benefited from the strong tracking association performance revealed by the AssocA (56.2% vs 52.9%). Compared with MOTRv2, CO-MOT slightly falls behind on TETA, but its AssocA (56.2%) is much better than MOTRv2 (51.9%).

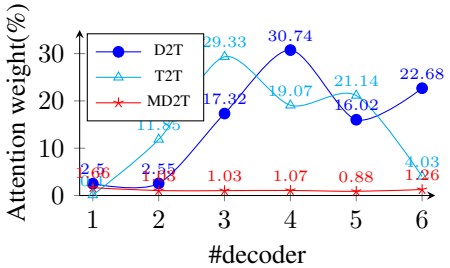

Figure 3: The attention weights between different types of queries on different decoders.

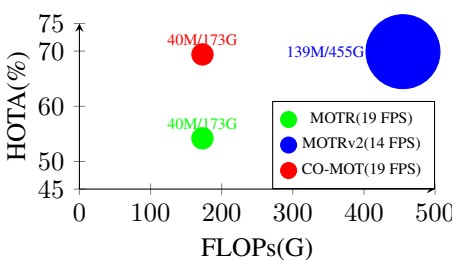

Figure 4: Efficiency comparison for CO-MOT and other end-to-end methods on the DanceTrack teset set.

**MOT17.** Table 2c shows the results of the MOT17 test set. Compared to the end-to-end methods, such as TrackFormer, MOTR, MeMOT( Cai et al. (2022)), MeMOTR, we still have significant improvement on HOTA. Although it is inferior to non-end-to-end methods such as P3AFormer( Zhao et al. (2022)), Deep OC-SORT, BoT-SORT, OC-SORT, ByteTrack, MAATrack( Stadler & Beyerer (2022)), GRTU( Wang et al. (2021b)), Unicorn( Yan et al. (2022a)), CorrTracker( Wang et al. (2021a)), we conjecture that the insufficient amount of MOT17 training data cannot be able to fully train a Transformer-based MOT model.

## 4.4 ABLATION STUDY

**Component Evaluation of CO-MOT.** Based on the results shown in Table 3a, we examine the impact of different components of the CO-MOT framework on tracking performance, as evaluated on the DanceTrack validation set. Through experimental analysis by combining various components, we achieve significant improvements over the baseline (61.8% vs 56.4%). By introducing the COLA strategy to the baseline (a), we observe an improvement of 3.8% on HOTA and 5.1% on AssA, without any additional computational cost. By incorporating the concept of shadow into the baseline (a), HOTA is improved by 2.6% and AssA is improved by 3.6%.

**COLA.** It is also evident from Table 3a that both COLA and Shadow have minimal impact on DetA (71.8% vs 73.5%), which is detection-related. However, they have a significant impact on AssA (44.6% vs 52.2%) and HOTA (56.4% vs 61.8%), which are more strongly related to tracking. On the surface, our method seems to help detection as it introduces more matching objects for detection, but it actually helps tracking.

To answer this question, we demonstrate the attention weights between detection and tracking queries in Figure 3. The horizontal and vertical axes denote the attention weights after self-attention between different types of queries on different decoder layers. These weights roughly indicate the contribution of one query to another. In our model, there are a total of 6 decoder layers. T2T represents the contribution of a tracking query to itself. D2T represents the contribution of a detection query predicting the same object to a tracking query. Two bounding boxes with an IOU greater than 0.7 are treated as the same object. MD2T represents the average contribution of all detection queries to a specific tracking query, which serves as a reference metric. Note that the normalized attention weights are with a sum of 1.

From Figure 3, it is evident that detection queries make a significant contribution (more than 15%) to their corresponding tracking queries in decoder layers where $L > 2$, even greater than the T2T for #4 and #6 decoders and much higher than the MD2T for all the decoders. This indicates that detection queries pass on the rich semantic information they represent to their corresponding tracking queries, which in turn can be utilized by the tracking queries to improve their tracking accuracy.

**Shadow Set.** Table 3c and Table 3b list ablation experiments related to three hyperparameters of shadow, which are the number of shadows, initialization method of shadows, and representative sampling strategies $\lambda$ and $\phi$. To choose the appropriate option for $\lambda$ and $\phi$, we first set $N_S$ to 5 and train the model only on the DanceTrack training set for 5 epochs using $I_{rand}$ without COLA. Then we try different combinations of $\lambda$ and $\phi$. It can be seen from Table 3b that the combination

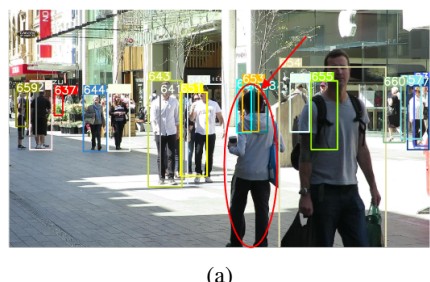 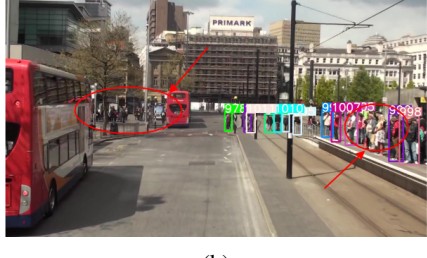

(a)                                                    (b)

Figure 5: Failed cases are often due to the failure to detect the target.

of $\lambda = max$ and $\phi = min$ yields the best results. That means we use the most challenging query in the set to train the model, leading to discriminative representation learning. To determine the initialization method, we also fix $N_S = 2$ with COLA and find that the best results are achieved using $I_{noise}$. For $I_{rand}$, there is a considerable variation between different shadows within the same set due to random initialization, making convergence difficult and resulting in inferior results. Finally, we try different values of $N_S$ and find that the best results are achieved when $N_S = 3$. When $N_S$ is too large, we observe that convergence becomes more difficult, and the results deteriorate.

### 4.5 EFFICIENCY COMPARISON

In Figure 4, efficiency comparisons on DanceTrack test dataset are made between CO-MOT and MOTR(v2). The horizontal axis represents FLOPs (G) and the vertical axis represents the HOTA metric. The size of the circles represents the number of parameters (M). It can be observed that our model achieves comparable HOTA (69.4% vs 69.9%) with MOTRv2 while maintaining similar FLOPs (173G) and number of parameters(40M) with MOTR. The runtime speed of CO-MOT is much faster (1.4×) than MOTRv2's. Thus, our approach is effective and efficient, which is friendly for deployment as it does not need an extra detector.

### 4.6 LIMITATIONS

Despite the introduction of COLA and Shadow, which improve the tracking effect of MOTR, the inherent data-hungry nature of the Transformer model means that there is not a significant improvement in smaller datasets like MOT17. As shown in Figure 5a, a prominently visible target has not been detected, but this issue has only been observed in the small MOT17 dataset. And due to the scale problem, the detection and tracking performance is poor for small and difficult targets in Figure 5b. In order to further improve the effect, it is necessary to increase the amount of training data or use a more powerful baseline such as DINO.

## 5 CONCLUSION

This paper proposes a method called CO-MOT to boost the performance of end-to-end Transformer-based MOT. We investigate the issues in the existing end-to-end MOT using Transformer and find that the label assignment can not fully explore the detection queries as detection and tracking queries are exclusive to each other. Thus, we introduce a coopetition alternative for training the intermediate decoders. Also, we develop a shadow set as units to augment the queries, mitigating the unbalanced training caused by the one-to-one matching strategy. Experimental results show that CO-MOT achieves significant performance gains on multiple datasets in an efficient manner. We believe that our method as a plugin significantly facilitates the research of end-to-end MOT using Transformer.

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
