# OpenReview forum: "CO-MOT: Boosting End-to-end Transformer-based Multi-Object Tracking via Coopetition Label Assignment and Shadow Sets"
_ICLR.cc/2024/Conference — Submitted to ICLR 2024_

### Official Review · Reviewer_gEi4 · 2023-10-19

**Soundness:** 2 fair
**Presentation:** 2 fair
**Contribution:** 2 fair
**Rating:** 3
**Confidence:** 5

**Summary:**

This paper is about using Transformer to perform multi-object tracking task. Authors introduce "coopetition alternative" strategy to train Transformer's decoders, also a "shadow set" is proposed to maintain a set of queries when tracking each object. Experiments are conducted on DanceTrack, BDD100K, MOT17 to compare with prior works.

**Strengths:**

1) This paper comes with a good motivation by first analyzing the drawbacks transformer based trackers, such as MOTR and MOTRv2, then try proposing some improvements based on that.

2) The idea of using a set of queries to track each object, is interesting.

3) Results on DanceTrack seems to be strong, e.g. it is comparable to MOTRv2 from Table 2(a) and it is better than a recent work MeMOTR, which is also based on Transformer.

**Weaknesses:**

1) The performance is not stable across different datasets. For instance, the tracker cannot even achieve comparable performance with existing trackers such as GTR/PA3Former/GRTU on MOT17, which is by far the most widely used and trusted benchmark within MOT community. While authors claim that MOT17 is rather a small dataset to train transformer-based trackers, other ways such as using synthetic dataset[a] could be one option.

 Also the proposed tracker performs worse than the major baseline MOTRv2, which is also transformer-based, in terms of IDF1 and HOTA. In this case, it is a bit hard to justify that authors are making a direction towards improving MOTRv2.

In fact, I would stay away with a rather heavier tracker (Transformer with multiple layers and many parameters) that gives even worse performance on MOT17 than light-weight tracking-by-detection/regression based trackers that require way less data to train, while still work better on major tracking benchmarks.

2) The current approach lacks some form of interpretability. For example, from this claim "COLA and TALA on the different decoders,", why different label assignment should be used in different decoders? What exactly happened inside the decoder part for Transformer-based MOT, is it solving data association exactly or approximately? It would be more advocated if authors could come up with a more insightful explanation for this, and it will also benefit readers.

3) For the idea of using shadow set, too many engineering tricks such as min, max, operations are used, it would be much better if authors could come up with some mathematical formulations as a technical sound approach to replace these simple heuristics, so as to better justify the use of this method. For example, IMHO, how about borrowing ideas from particle filter, to use importance sampling to maintain a set of queries to track each object?

References:
[a] Motsynth: How can synthetic data help pedestrian detection and tracking? ICCV2021

**Questions:**

I see in Table 3(a) that adding COLA and shadow set both improves performance than not having them, but this is only tested on DanceTrack, would the COLA and shadow set also work better than not having them at all, on MOT17 validation set?

---

> ### Author Response · Authors · 2023-11-20
>
> We thank the reviewer gEi4 for the insightful comments and suggestions. We detailedly respond to each comment in the following.
>
> **Response to Weaknesses 1:**
>
> 1.1  Due to the limited time of the rebuttal, we will supplement the results adding synthetic dataset.
>
> 1.2 MOTRv2 is transformer-based, but due to the addition of YOLOX preprocessing, it has a larger computational load and is not a pure End-to-end method. The original intention of this paper is to maintain the End-to-end structure and computational load of MOTR and achieve performance comparable to MOTRv2. The strong performance in Table 2 has well supported our motivation on multiple datasets.
>
> 1.3  The table below lists the performance of known End-to-end methods on the MOT17 test. It can be seen that End-to-end tracking has been prevalent in the past several years, and its performance has gradually improved. Currently, End-to-end MOT methods are still in the early stages of development, and we believe that there will be significant improvements in the future. The value of a method should not be denied just because it has not achieved state-of-the-art performance on a particular dataset. For example, although DETR and Deformable DETR have lower performance on the COCO dataset compared to other non-Transformer detection methods, their contributions to subsequent methods cannot be ignored.
>
> | |Source|HOTA| DetA| AssA| MOTA| IDF1|
> |---|---|---|---|---|---|---|
> |MeMOT          | CVPR’22|56.9| - |55.2| 72.5| 69.0|
> |TrackFormer    |CVPR’22| - | - | - |65.0| 63.9|
> |***MQT***            |arXiv’22|  - | - | - |66.5| 65.2|
> |***DNMOT***          |arXiv’23|58.0|-|-|75.6|68.1|
> |MOTR           |ECCV’22|57.8| 60.3| 55.7| 73.4| 68.6|
> |MeMOTR         |ICCV’23| 58.8| 59.6 |58.4| 72.8| 71.5|
> |***MOTRv3 (ConvNeXt-B)***| arXiv’23 |60.1| 58.7|62.1|75.9| 72.4|
> |CO-MOT         |-      | **60.1**| 59.5| 60.6| 72.6| **72.7**|
>
> **Table 2 (c)**:  Comparison to End-to-end methods on the MOT17 test dataset.  The methods with ***bold italic*** fonts are newly introduced. CO-MOT has achieved the highest HOTA(60.1). Although the HOTA performance of MOTRv3 is the same as CO-MOT, it uses a more powerful backbone (ConvNeXt-B).
>
> **Response to Weaknesses 2:**
>
> The following table studies the performance impact of COLA inserting different decoder layers. l<0 means that COLA is not used, and all 6 layers of the decoder use TALA. l<2 means that the first layer of the 6-layer decoder uses COLA. It can be seen that deploying COLA in more decoder layers leads to better HOTA. As shown in Figure 3, during training in MOTR, TALA forcibly matches the tracking query with the corresponding track object, even if the current prediction of the tracking query is far from the target. This strategy makes it more difficult for the network to converge compared to the optimal solution of the Hungarian matching, which in turn makes it difficult for the track query to reach the optimal solution. This can also be seen in Figure 1 in the motivation section. Introducing COLA can improve this convergence. The detection query recalls the tracked targets through Hungarian matching and then passes the relevant information to the track query through the attention mechanism of the Transformer. With the prior knowledge from the detection query, the convergence of the track query becomes easier, and the target will be consistently assigned the same identity.
>
> |  l |HOTA| DetA| AssA| MOTA| IDF1|
> |---|---|---|---|---|---|
> |l<0|58.6 | 72.3 |47.7 |80.1 |58.9|
> |l<2|59.0| 72.9 |   48.1 |81.2 |59.8 |
> |l<4|59.6 |   74.3  |  48.0 |82.8| 60.5 |
> |l<6|59.9 |   73.2 |   49.3| 81.3| 60.9|
>
> **Table 8**:  The performance metrics of COLA inserting different numbers of decoder layers.
>
> **Response to Weaknesses 3:**
>
> Thank the reviewer for the provided advice to make this work more technically solid. "min" and "max" used to select the samples for training are heuristic but effective. More exploration such as importance sampling for sample selection will be conducted in our future work.
>
> **Response to Question 1:**
>
> The following two tables respectively show the ablation experiments of inserting COLA and Shadow set into TrackFormer and MOTR. As components are added, the tracking performance gradually improves.
>
> | |COLA|	Shadow|	HOTA|
> |---|---|---|---|
> |(a) TrackFormer| | | 41.4 |
> |(b) ours | ✓| | 47.8(+6.4) |
> |(c) ours  |✓ | ✓ |50.7(+9.3) |
>
> **Table 4**:  The ablation experiments of applying CO-MOT to TrackFormer on the DanceTrack validation set.
>
> | |COLA|	Shadow|	HOTA of Dancetrack|  HOTA of MOT17|
> |---|---|---|---|---|
> |(a) MOTR| | |  56.4| 57.8|
> |(b)| ✓| | 60.2(+3.8)| 58.5(+0.7)|
> |(c)| ✓ |✓ | 61.8(+5.4)| 60.1(+2.3)|
>
> **Table 9**:  The ablation experiments of applying CO-MOT to MOTR on the DanceTrack validation set and MOT17 test.

---

> ### Comment · Reviewer_gEi4 · 2023-11-21
>
> Thanks for the author for the feedback. I have carefully read the rebuttals.
>
> 1. Regarding "End-to-end MOT methods are still in the early stages of development, and we believe that there will be significant improvements in the future. The value of a method should not be denied just because it has not achieved state-of-the-art performance on a particular dataset..." I do agree that not achieving SOTA should not be the reason to reject a submission. But, DETR is the first work to make Transformers work for object detection and thus is a $\textbf{0-1}$ work, so no one will criticize its inferior performance compared to non-Transformer based detectors and it has influenced lots of follow-up works.  However, the current submission is nothing like DETR, as lots of Transformer based trackers have already been published, e.g. TrackFormer/MOTR/MOTRv2, therefore the current work is not $\textbf{0-1}$ work like DETR but $\textbf{1-2, 2-3}$ $\textbf{incremental}$ work, it is just about "climbing mountains" yet the performance is not even better than non-Transformer based trackers, though I'm not criticizing it not being SOTA. Making it impossible to justify the motivation as, sadly I didn't see novel insights nor strong results.
>
> 2. I do agree with the argument that " More exploration such as importance sampling for sample selection will be conducted in our future work". I hope the author could improve towards a better formulation for this part in the future work.
>
> I think for conferences like NeurIPs/ICLR, we should have more expectations for submissions. I therefore continue with my rating and I hope author would continue working on improving it towards a stronger submission.

---

### Official Review · Reviewer_oLii · 2023-10-24

**Soundness:** 3 good
**Presentation:** 3 good
**Contribution:** 2 fair
**Rating:** 6
**Confidence:** 5

**Summary:**

This paper proposes an end-to-end Transformer-based MOT method. It introduces a coopetition alternative for training the intermediate decoders, and also develops a shadow set as units to augment the queries. Experimental results on some public datasets demonstrate the effectiveness of the proposed method.

**Strengths:**

1. This paper is well written and organized. The S-COLA is also clearly explained.
2. Experimental results on some public datasets demonstrate the effectiveness of the proposed method.

**Weaknesses:**

1. For the tracking queries and the detection queries, they both self-attend each other and then cross-attend the image feature tokens. Intuitively, all the queries can perceive the whole image, and it is not good enough to split them into two parts, which may degrade the detection results.

2.  Why not evaluate the proposed method on the crowded dataset MOT20? In crowd scenes, the tracking query has the similar semantic information with the detection query, and may fail to assign correct label in crowd scenes.

3. In Table 1, the authors point out that the detection result of MOTR can be improved by removing the tracking queries. How about the proposed CO-MOT? If CO-MOT is only used as a detection task, and whether the tracking performance can be further improved?

4. It is better to evaluate the effect of the number of decoders L on the tracking performance.

5. Pease reorganize the formats of the references, and make them consistent.

**Questions:**

See the Weaknesses.

---

> ### Author Response · Authors · 2023-11-20
>
> We thank the reviewer oLii for the insightful comments and suggestions. We detailedly respond to each comment in the following.
>
> **Response to Weaknesses 1:**
>
> In the multi-object tracking (MOT) task, it is necessary to keep tracking the targets that are tracked in the previous frame, while also detecting newborns that appear in the current frame. The tracking queries and detection queries play different roles in these tasks. During training, they are supervised by different supervision, where the detection queries use Hungarian matching, and the tracking queries use TALA. During inference, the tracking queries and detection queries are concatenated together and indiscriminately passed through multiple Transformer decoders. Thus, all the queries can perceive the whole image, as the reviewer points out, but both tracking and detection queries must exist for MOT.
>
> **Response to Weaknesses 2:**
>
> As the End-to-End solution has just emerged in the past year, there are not many methods evaluated on MOT20 that we could find. Here are the ones in Table 2 (d). Notably, our approach achieves 57.5% HOTA, which is the state-of-the-art in End-to-end tracking methods.
>
> | |Venue |HOTA| DetA| AssA| MOTA| IDF1|
> |---|---|---|---|---|---|---|
> |MeMOT | CVPR’22 | 54.1| \ | 55.0 | 63.7| 66.1|
> |TrackFormer|CVPR’22| 54.7| \ | \ |  68.6| 65.7 |
> |CO-MOT| -| **57.5** | 50.5 | **65.7** | 60.1 |	**70.5**|
>
> **Table 2 (d)**: Comparison to existing methods  on the MOT20 test dataset.
>
> **Response to Weaknesses 3:**
>
> The following table lists the mAP indicators, which are consistent with the DetA indicators in Table 2(a). The detection performance of CO-MOT is far superior to MOTR but slightly inferior to MOTRv2.
>
> Tracking tasks not only include detecting new targets but also need to track targets that have already been tracked, maintaining the identity of the targets between consecutive frames. Therefore, CO-MOT cannot be used only as a detection task, otherwise, it would not be able to associate targets between previous and subsequent frames.
>
> | method | mAP|
> |---|---|
> |(a) MOTR | 42.5 |
> |(b) MOTRv2|  70.7 |
> |(c) CO-MOT|  69.1|
>
> **Table 6**:  The detection performance (mAP) of MOTR (v2) and CO-MOT on DanceTrack validation dataset.
>
> **Response to Weaknesses 4:**
>
> We ablate the number of decoders and illustrate the results in Table 7. The results show that the HOTA consistently improves as the number of decoders increases, while other metrics significantly increase or are comparable. To make a fair comparison with MOTR and consider the efficiency, we choose six decoders by default in all the experiments.
>
> |  L |HOTA| DetA| AssA| MOTA| IDF1|
> |---|---|---|---|---|---|
> |L=2|49.0  |  51.9 |   46.6 | 53.8 |   53.2|
> |L=4|58.2  | 71.7  |  47.5 | 80.8| 60.7|
> |L=6|61.0  |  73.9  |  50.6| 82.7 |62.6|
> |L=8|61.5 |  77.0 |   49.3 | 86.6| 62.6 |
>
> **Table 7**:  Ablation study on the number of decoders L. As L increases, the computational load increases, and the HOTA tracking performance improves, but the growth rate of HOTA significantly decreases.
>
> **Response to Weaknesses 5:**
>
> We reorganize the formats of the references according to the official regulations in the manuscript.

---

### Official Review · Reviewer_NAsG · 2023-10-31

**Soundness:** 4 excellent
**Presentation:** 3 good
**Contribution:** 4 excellent
**Rating:** 8
**Confidence:** 4

**Summary:**

The paper works on an end-to-end Multi-Object Tracking problem via a query-based Transformer, which assumes that the existing label assignment leads to scarce positive samples for detecting newborns. With this limitation in training, this paper observes a tracking terminal without renewal or re-initialization compared to other tracking-by-detection methods. To remedy this issue, this work presents Co-MOT by matching detection queries to all the tracked and untracked targets. Additionally, each query is augmented by a set with limited disturbance to itself. CO-MOT, with extra costs, achieves 69.4 HOTA on DanceTrack and 52.8 TETA on BDD100K, making it 1.4 times faster than the previous approach MOTRv2.

**Strengths:**

Deep observation of the existing approaches: This paper provides qualitative results in Figure 1 and quantitative results in Table 1 to bring the readers to what has been done to the existing e2e-MOT via Transformer. The bipartite matching label assignment in the previous approaches has a limitation regarding the video task, compared to the image task, such as object detection. From this perspective, this work involves the inherent property of the e2e-MOT via Transformer and offers a promising solution to the limitation. The superior performance of the proposed method is inspiring, which only requires fewer FLOPS of MOTRv2 to achieve a similar performance. I reckon this work reveals the core problem of the Transformer-based e2e-MOT.

Simple and effective solutions: The paper reveals that the target assignment on detection queries is scarce due to the exclusive tracking queries, leading to unbalanced training. The proposed coopetition label assignment solves this problem by allowing the detection queries to match all the targets in the intermediate decoders except the last decoder. This simple solution can effectively recall newborn detection in the intermediate decoders while keeping standard MOT in the final outputs.  In addition, the shadow concept is introduced to augment each query in a feature-augmentation manner. A bulk of ablations are provided to demonstrate the effectiveness of each proposed solution.

**Weaknesses:**

Missing experiments on more models: In Section 3.2, this paper mentions that we revisit the Tracking Aware Label Assignment (TALA) used to train end-to-end Transformers such as MOTR and TrackFormer for MOT. However, all the ablations are conducted on MOTR. I cannot find evidence that COLA and shadow work on other models like TrackFormer. The authors should provide results of TrackFormer on two datasets at least to make the proposed solutions more solid.

Comparison of failure cases: The failure cases shown in Figure 5 are the common cases of all the Transformer-based approaches.  The authors should provide the failure cases of previous approaches and denote which kind of case has been solved by CO-MOT.

**Questions:**

In Table 2, the authors categorize all the methods into Non-End-to-end and End-to-end. What are the exact definitions of these two categories? Also, it will be better to list all the methods using Transformer.

In Figure 4, do the FLOPs of the MOTR include the YOLOX detector? What are the specific numbers of parameters for different approaches? The specific number of parameters should be indicated in Figure 4. For example, the number can be placed in circles.

---

> ### Author Response · Authors · 2023-11-20
>
> We thank the reviewer NAsG for the insightful comments and suggestions. We detailedly respond to each comment in the following.
>
> **Response to Weaknesses 1:**
>
> In the supplementary materials, we provide the results of COLA and Shadow Set on Trackformer. We do not make any modifications to the TrackFormer framework and used the default hyperparameters provided by TrackFormer. The result shown as follows demonstrates the effectiveness of COLA (+6.4\%) and Shadow Set (+9.3\%) when applied to TrackFormer. Codes will be released upon acceptance.
>
> | |COLA|	Shadow|	HOTA|
> |---|---|---|---|
> |(a) TrackFormer| | | 41.4 |
> |(b) ours | ✓ | | 47.8 (+6.4) |
> |(c) ours  |✓ | ✓ |50.7 (+9.3) |
>
> **Table 4**:  The ablation experiments of applying CO-MOT to TrackFormer on the DanceTrack validation set. As components are added, the tracking performance improves gradually.
>
> **Response to Weaknesses 2:**
>
> The Figure 7 illustrates the failure cases of MOTR and denotes which kind of case has been solved by CO-MOT. MOTR has poor detection and tracking performance. First, as shown in Figure 7 (a), it fails to detect the person in time under the extreme case of bending over. Second, as shown in Figure  7 (b), due to the tiny visual features when a person stretches out their hand, the detection box is inaccurate, and the model misidentifies it as multiple people. Third, as shown in Figure  7 (c), the tracking identity switches after the human body is obscured. However, all of the above cases can be solved by CO-MOT, showing the extraordinary performance of MOT.
>
> **Response to Question 1:**
>
> End-to-end tracking [1,2,3,4] is defined as inputting contiguous video frames and directly outputting the boxes and association information without any pre- or post-processing (such as separate detectors, Reid feature extraction, or IOU match). So methods such as MOTRv2, ByteTrack, TransTrack, and OC-SORT belong to Non-End-to-end tracking. On the contrary, MOTR, MeMOTR, MeMOT does indeed belong to an end-to-end approach. We append more Transformer-based methods for comparison on MOT17 in Table 2 (c). The methods with ***bold italic*** fonts are newly introduced.
>
> ||Source|HOTA| DetA| AssA| MOTA| IDF1|
> |---|---|---|---|---|---|---|
> |TransTrack     |arXiv’20| 54.1| 61.6| 47.9| 74.5| 63.9|
> |P3AFormer      | ECCV’22| - | - | - |81.2| 78.1|
> | ***DiffusionTrack*** | arXiv’23 | 60.8| 63.2| 58.8| 77.9| 73.8|
> |MOTRv2         |CVPR’23|62.0| 63.8| 60.6| 78.6| 75.0|
> |---|---|---|---|---|---|---|
> |MeMOT          | CVPR’22|56.9| - |55.2| 72.5| 69.0|
> |TrackFormer    |CVPR’22| - | - | - |65.0| 63.9|
> |***MQT***            |arXiv’22|  - | - | - |66.5| 65.2|
> |***DNMOT***          |arXiv’23|58.0|-|-|75.6|68.1|
> |MOTR           |ECCV’22|57.8| 60.3| 55.7| 73.4| 68.6|
> |MeMOTR         |ICCV’23| 58.8| 59.6 |58.4| 72.8| 71.5|
> |***MOTRv3 (ConvNeXt-B)***| arXiv’23 |60.1| 58.7|62.1|75.9| 72.4|
> |CO-MOT         |-      | **60.1**| 59.5| 60.6| 72.6| **72.7**|
>
> **Table 2 (c)**:  Comparison to existing methods using Transformer  on the MOT17 test dataset.  The last few lines are all end-to-end tracking methods, among which CO-MOT has achieved the highest HOTA. Although the HOTA performance of MOTRv3 is the same as CO-MOT, it uses a more powerful backbone (ConvNeXt-B).
>
> **Response to Question 2:**
>
> We show the specific numbers in the following table and indicate them in Figure 4.
>
> | | params| FLOPs| Inference FPS| HOTA|
> |---|---|---|---|---|
> |MOTR| 40M | 173G| 19| 54.2|
> |MOTRv2| 139.1M | 454.9G | 14| 69.9|
> |CO-MOT| 40M | 173G| 19 | 69.4|
>
> **Table 5**: The parameters, model size, and FLOPs of MOTR, MOTRv2, and CO-MOT.
>
> References:
>
> [1] F. Zeng et.al. Motr: End-to-end multiple-object tracking with transformer. ECCV 2022
>
> [2] Y. Zhang et.al. Motrv2: Bootstrapping end-to-end multi-object tracking by pretrained object detectors. CVPR2023
>
> [3] J. Cai et.al. Memot: multi-object tracking with memory. CVPR2022
>
> [4] T. Meinhardt et.al. Trackformer: Multi-object tracking with transformers. CVPR2022
>
> [5] Yu et.al. MOTRv3: Release-Fetch Supervision for End-to-End Multi-Object Tracking. https://arxiv.org/abs/2305.14298

---

### Official Review · Reviewer_kcLe · 2023-10-31

**Soundness:** 3 good
**Presentation:** 2 fair
**Contribution:** 2 fair
**Rating:** 6
**Confidence:** 2

**Summary:**

The paper introduces CO-MOT, a method that enhances the performance of end-to-end Transformer-based MOT. It addresses issues in the existing framework where label assignment fails to fully utilize detection queries due to the exclusive nature of detection and tracking queries. To overcome this, a coopetition alternative is proposed for training intermediate decoders. Additionally, a shadow set is developed to augment queries and mitigate training imbalances caused by one-to-one matching. The experiments are conducted on MOT17, BDD100k and DanceTrack.

**Strengths:**

+ This paper mentioned that there is a large amount of post-processing in non-end-to-end methods in the MOT field, which leads to difficulties in parameter tuning in application. In this paper, some improvements have been made based on existing end-to-end methods, resulting in certain speed and accuracy improvements.

+ This paper analyzes the shortcomings of existing e2e-MOT models and proposes a new network design strategy based on the analyses.

**Weaknesses:**

- The layout of the article is quite chaotic. Figure 1 is used as an example in the method introduction, but it is too far from the text information. For another example, in Section 3.2 of the introduction of strategies and methods, formulas are mixed in a large paragraph without a comprehensive formula or image description, making it difficult for readers to understand. The reviewer must refer to the paper of the baseline tracker MOTR for a comprehensive understanding. The authors should re-organize the method introduction and include a detailed figure to highlight the technical improvements of the new approach, including both the label assignment and shadow set.

- The comparisons with more tracking approaches on the BDD100k or DanceTrack datasets are required to demonstrate the performance improvement.

**Questions:**

Please see the Weaknesses section.

---

> ### Author Response · Authors · 2023-11-20
>
> We thank the reviewer kcLe for the insightful comments and suggestions. We detailedly respond to each comment in the following.
>
> **Response to Weaknesses 1:**
>
> Thank you for your constructive feedback on the paper organization. We have updated the manuscript according to your suggestions to improve its readability and comprehensibility. For example, we move Figure 1 near the method section so that readers can refer to it more conveniently. Additionally, we demonstrate the symbols of Section 3.2 in Figure 2. To highlight the technical improvements of the new approach, we add a detailed figure of MOTR (v2) and CO-MOT in Figure 6. Note that the stacked circles and squares denote the shadow sets.
>
> **Response to Weaknesses 2:**
>
> We demonstrate tracking methods as many as possible in the supplementary materials, which are evaluated on DanceTrack. The methods with ***bold italic*** fonts are newly introduced. For BDD100K, we have already shown all the methods evaluated in Table 2(b) so that no more methods are introduced.
>
>
> | |Source | HOTA   | DetA | AssA | MOTA | IDF1 |
> |---|---|---|---|---|---|---|
> ***SORT***          |ICIP’16 |47.9| 72.0 |31.2| 91.8 | 50.8|
> ***CenterTrack***    | ECCV’20|41.8| 78.1| 22.6| 86.8| 35.7|
> TransTrack     |arXiv’20| 45.5| 75.9| 27.5| 88.4| 45.2|
> ***FairMOT***        |IJCV’21| 39.7| 66.7| 23.8| 82.2| 40.8|
> QDTrack        | CVPR’21|54.2| 80.1| 36.8| 87.7| 50.4|
> ***TraDeS***         | CVPR’21| 43.3| 74.5| 25.4| 86.2| 41.2|
> ByteTrack      | ECCV’22| 47.7| 71.0| 32.1| 89.6| 53.9|
> OC-SORT        | CVPR’23| 55.1| 80.3| 38.3| 92.0| 54.6|
> ***GTR***            |CVPR’22| 48.0| 72.5| 31.9| 84.7| 50.3|
> ***C-BIoU***         | WACV’23 |60.6| 81.3| 45.4| 91.6| 61.6|
> ***MT IoT***         | arXiv’22| 66.7| 84.1| 53.0| 94.0| 70.6|
> DiffusionTrack |arXiv’23|52.4|82.2 |33.5| 89.3| 47.5|
> MOTRv2+        | CVPR’23| 69.9| 83.0| 59.0| 91.9| 71.7|
> |---|---|---|---|---|---|---|
> MOTR           |ECCV’22| 54.2| 73.5| 40.2| 79.7| 51.5|
>  ***DNMOT***          |arXiv’23|53.5|-|-|89.1|49.7|
> MeMOTR         |ICCV’23| 63.4| 77.0| 52.3| 85.4| 65.5|
> MeMOTR*        |ICCV’23| 68.5| 80.5| 58.4| 89.9| 71.2|
> ***MOTRv3 (Res50)***  |arXiv’23|68.3| -|-|91.7| 70.1|
> CO-MOT         |-      | 65.3| 80.1| 53.5| 89.3| 66.5|
> CO-MOT+        |-      | **69.4**| **82.1**| **58.9**| 91.2| **71.9**|
>
> **Table 2 (a)**:  Comparison to existing methods on the DanceTrack test set. The last few lines are all end-to-end tracking methods, among which CO-MOT has achieved the highest HOTA.

---

### Meta-Review · Area_Chair_kwSK · 2023-12-08

**Metareview:**

The paper initially had diverging reviews (3, 6, 6, 8). The major concerns of the reviewers were:

1. presentation issues, need to refer to baseline MOTR paper [kcLe]
2. Needs more comparison on BDD100k and DanceTrack [kcLe]
3. Missing evidence that COLA/Shadows works on other models like Trackformer. [NAsG]
4. Provide failures cases specific to previous approaches that were solved by CO-MOT [NAsG]
5. Table 2 presentation issues [NAsG]
6. Fig 4 - show the number of parameters, FLOPS of YOLOX included in MOTR? [NAsG]
7. Is it necessary to split tracking/detection queries? [oLii]
8. Missing evaluation on MOT20. [oLii]
9. What if the tracking queries are removed in CO-MOT. Does detection result improve similar to MOTR? [oLii]
10. effect of the number of decoders L on tracking performance [oLii]
11. Performance improvement not consistent across different datasets [gEi4]
12. Performance worse than MOTRv2 in terms of IDF1 and HOTA [gEi4]
13. Lack of interpretability of proposed method [gEi4]
14. Lack of mathematical formulation for shadow sets -- too many engineering tricks or heuristics [gEi4]
15. missing ablation study on w/ and w/o COLA/shadow set on MOT17 validation set. [gEi4]

The authors wrote a response to address these concerns. During the discussion, Rev gEi4 was still concerned about the novelty being incremental, since the proposed method mainly works to enhance the detector for MOT and not the data association, and the proposed methods used are heuristic and without strong mathematical formulations or justifications. Rev oLii agreed with Rev gEi4 about the novelty.  Meanwhile, Rev oLii was not convinced with the MOT20 results, since the proposed method seems worse than two-stage methods. Rev gEi4 also agreed that the MOT20.  In contrast, Rev NAsG thought that the method was able to rectify the shortcomings of MOTRv2 by introducing efficient label assignment and shadow sets, achieving efficiency and competitive performance.  Rev NAsG also thought that the inferior performance on MOT17/20 was due to the small scale of the datasets, and the proposed method showed its advantage on BDD100K and DanceTrack, which are larger datasets.

Overall, the AC agrees with the concerns of Rev gEi4 and oLii.  The methodology is heuristic and not strongly mathematically motivated or analyzed. Thus, the proposed work lacks methodological depth or broad applicability. Rev gEi4 offers a good suggestion to improve the formulation of ShadowSets by borrowing from e.g., particle filters or importance sampling. Overall the results are promising, though. Thus, the AC recommends reject.

**Justification For Why Not Higher Score:**

While the results are promising (better than previous end-to-end trackers, but not better than transformer trackers), the methodology is incremental and lacks mathematical formulations/justifications. It will be a very narrow audience in ICLR.

**Justification For Why Not Lower Score:**

n/a

---

### Decision · Program_Chairs · 2024-01-16

Reject